# Pill Box Text Identification Using DBNet-CRNN

**DOI:** 10.3390/ijerph20053881

**Published:** 2023-02-22

**Authors:** Liuqing Xiang, Hanyun Wen, Ming Zhao

**Affiliations:** School of Computer Science, Yangtze University, Jingzhou 434025, China

**Keywords:** deep learning, text recognition, natural scenes, convolutional recurrent neural network, image processing

## Abstract

The recognition process of natural scenes is complicated at present, and images themselves may be complex owing to the special features of natural scenes. In this study, we use the detection and recognition of pill box text as an application scenario and design a deep-learning-based text detection algorithm for such natural scenes. We propose an end-to-end graphical text detection and recognition model and implement a detection system based on the B/S research application for pill box recognition, which uses DBNet as the text detection framework and a convolutional recurrent neural network (CRNN) as the text recognition framework. No prior image preprocessing is required in the detection and recognition processes. The recognition result from the back-end is returned to the front-end display. Compared with traditional methods, this recognition process reduces the complexity of preprocessing prior to image detection and improves the simplicity of the model application. Experiments on the detection and recognition of 100 pill boxes demonstrate that the proposed method achieves better accuracy in text localization and recognition results than the previous CTPN + CRNN method. The proposed method is significantly more accurate and easier to use than the traditional approach in terms of both training and recognition processes.

## 1. Introduction

With the development of computer technology, artificial intelligence has made considerable progress in recent years. Optical character recognition (OCR) [1], which is a field of computer science, has also gradually appeared in various industries. OCR has traditionally involved the processing and analysis of images to recognize specific information. However, with continual technological developments, OCR is often defined as an all-image text detection and recognition technology in a broad sense (graphic recognition technology), which includes both traditional OCR recognition technology and scene text recognition [2].

Traditional OCR recognition may yield distortions, folds, mutilations, and blurring, and in some cases, the text in the image is distorted. Moreover, because most document images are of regular tables, a large-scale and effective preprocessing process, that is, the use of image preprocessing technology, is required in advance to achieve a high level of text detection and recognition. The preprocessing may include image segmentation, image rotation correction, line detection, image matching, text outline extraction, and local segmentation.

Image segmentation is the process of dividing an image into several specific regions with unique properties in which each region is a continuous set of pixels. It is a key step in image processing and analysis. Commonly used segmentation methods are mainly divided into the following categories [3]: threshold-based, region-based, and edge-based segmentation methods. In recent years, researchers have continuously improved the original image segmentation methods. Several new theories and methods from other disciplines have been used for image segmentation and many new segmentation methods have been proposed, such as those based on specific theories, cluster analysis [4,5], fuzzy set theory [6], wavelet transform, and neural networks [7]. However, as no general theory has been developed thus far, most existing segmentation algorithms are problem-specific and do not provide a general segmentation method that is suitable for all images. In practical applications, multiple segmentation algorithms are generally used together effectively to achieve better image segmentation results.

Image rotation correction is necessary because images that are obtained via printer scanning or cell phone photography may be rotated; for example, a rotated bill image will seriously affect the positioning of the rows and columns in the bill image layout, thereby increasing the difficulty of bill layout analysis. Therefore, the use of an effective rotation correction method [8,9,10] to correct bill images can significantly improve the performance of the bill recognition system. 

The line detection technique refers to the need for the straight line detection of images in certain recognition processing scenarios. The accuracy of straight line detection is decisive for the OCR localization of form-based bills. Commonly used methods [11,12,13] include projection transform, Hough transform, chain code, and tour, among which the Hough transform algorithm and its improvements have exhibited improved practicality [14]. Furthermore, vectorization algorithms [15,16] offer a wide range of applications in the straight line detection process. 

Image matching techniques can be applied to character localization and stamp detection. The position of the region of interest can be retrieved to locate the corresponding character position indirectly with the aid of image matching algorithms. Many matching methods have been proposed to improve the matching speed and performance; for example, the mutual correlation detection image matching algorithm [17], similarity measure matching method [18], adaptive mapping matching method [19], scale-invariant feature transform matching algorithm [20], and genetic-algorithm-based image matching method [21]. 

Character outline extraction and local segmentation techniques are required to segment the characters completely prior to character recognition. Common methods include threshold segmentation [22], morphological segmentation [23], and model-based segmentation algorithms. The general character segmentation algorithm first performs vertical projection, then analyzes the character outline, and finally segments the characters according to a predetermined character pitch or size. This approach is mainly divided into the following two categories: row segmentation [24] and isolated image decomposition [25].

With the development of deep learning technology, the complex preprocessing of traditional OCR is gradually being replaced with deep learning with simple steps [26]. The OCR process mainly includes two models: a text detection model and a text recognition model. At the inference stage, the two models are combined to build the entire graphic recognition system. End-to-end graphic detection and recognition networks have emerged in recent years. In this approach, during the training phase, the input to the model contains the image to be trained, text content in the image, and coordinates corresponding to the text. Following the inference phase, the original image is predicted directly using the end-to-end model with text content information. Xuebo Liu et al. proposed an end-to-end network for FOTS, which saves time and learns more image features than the traditional approach [27]. Christian Bartz et al. proposed an STN-OCR end-to-end network, which improves the accuracy of the recognition phase by embedding a spatial transformation network in the detection process to perform affine transformation on the input image [28]. Siyang Qin et al. proposed an unconstrained end-to-end network to simplify the recognition process by reducing the detection of arbitrarily shaped text to an instance segmentation problem without the need to perform text region correction in advance [29].

This method is simpler to train and more efficient than the traditional OCR recognition approach, and it can enable easier deployment through model optimization.

With the implementation of information technology in pharmaceutical systems, more and more complex and redundant tasks can be gradually improved by means of information technology. Among them, the entry of drug box information is a time-consuming and labor-intensive task that is also error-prone. Based on the excellent recognition capabilities of end-to-end deep learning networks, they can be used to reduce the complexity of this task while improving the accuracy of entry. In this paper, we propose and design a deep-learning-based end-to-end scene text recognition algorithm for the task of recognizing pill box text.

The contributions of this study are summarized as follows:

(1) A deep-learning-based scene text recognition algorithm was designed to recognize medicine names from the text of pill boxes. Considering the complexity of pill box images, this algorithm is more convenient for training than traditional OCR algorithms, and it does not require complicated and redundant image preprocessing. Thus, the ease of use of the entire algorithm is improved.

(2) DBNet, which is superior to other models in detecting text regions in scene text recognition, was used as the image text region detection model. Thus, the accuracy of the text analysis was improved.

(3) The B/S-based detection architecture, which is convenient for text content recognition, was used to reduce the learning cost of the users in the scenes, thereby significantly improving the ease of use of the model. 

This paper is organized in the following manner: Section 2 discusses related work, primarily in the area of deep learning and neural networks. This is followed by a description of several deep-learning-driven graphical neural networks that differ from traditional approaches to graphical recognition. We describe the proposed deep learning-based method for recognizing pill boxes in Section 3. The concept and principle of the method are also discussed in detail. The experimentation process for deep-learning-based pill box recognition is elaborated on in Section 4, along with the specific network training process and the specific training results. In addition to the architecture design of the recognition system, the final experimental results are presented. Lastly, the experimental results presented in Section 4 are analyzed and the next research direction is determined.

## 2. Related Work

### 2.1. Deep Learning

Deep learning [30], which was proposed in 2006, is a new branch of machine learning. Unlike traditional shallow models, deep learning models are relatively structurally complex and do not rely on manually acquired text features. This enables the direct learning and modeling of textual content. Furthermore, deep learning clarifies the importance of feature learning and makes natural language processing easy by transforming the feature representation of the samples in the original space into a more meaningful feature space through layer-by-layer feature transformation.

#### 2.1.1. Recurrent Neural Networks

Recurrent neural networks (RNNs) are efficient for processing data with sequential characteristics as they can mine both the temporal and semantic information in the data. This feature of RNNs has led researchers to focus on problems in natural language processing fields.

The basic structure of an RNN consists of an input layer, a hidden layer, and an output layer. In the RNN, each computation feeds the output of the current layer into the next layer. This layer computes the output along with the input of the previous layer.

Although RNNs exhibit superior performance in processing sequential data, they have less impact on long-term memory and cannot cope with very long inputs.

#### 2.1.2. Long- and Short-Term Memory Neural Networks

Long short-term memory (LSTM) [31] is an RNN that is better at training long-sequence tasks than the common RNN. LSTM consists of three phases. The forgetting phase selectively deletes the data from the previous node, the selective memory phase remembers the inputs of the current phase selectively, and the output phase determines the output of the current phase. The transmission state of LSTM is controlled by the gate state.

#### 2.1.3. Convolutional Neural Networks

The main features of a convolutional neural network (CNN) are weight sharing and local connectivity [32]. CNNs are generally used as feedforward neural networks to process image data. In a CNN, the input images are scanned using a convolutional kernel. The number in the kernel is the weight and each position of the image is scanned using the same kernel, which is known as weight sharing. Each neuron only needs to perceive the local information of the current network layer for images. Global information can be obtained by combining the local information from the lower layers. The local connectivity can improve the training efficiency by reducing the number of parameters.

Unlike in image processing, the input to the CNN needs to be transformed into a matrix representation of sentences or documents in natural language processing. Each row of the matrix corresponds to a vector representation of an element, which may be either a word or a character. These vectors can be obtained in the form of word embeddings or one-hot encodings. The convolutional layer is an essential component of a CNN. Each node of the convolutional layer corresponds to the local information of the previous layer of the network using convolutional operations. This enables focus to be placed on different features of the input image or text. Different features can be extracted from a text sequence by changing the size of the convolution kernel. Similar to the operation of the convolutional layer, the pooling layer uses convolution kernels to extract the features.

However, the pooling layer only considers the maximum or average of the convolution kernels in the corresponding position.

The pooling layer continuously reduces the number of parameters, which increases the computational speed, reduces the computational effort, and controls overfitting to a certain extent. A fully connected layer is generally added following the convolution and pooling layers to minimize the dimensionality and retain useful information. This is a result of vector convolution and morphological combination. Following feature extraction, the data can be transferred to the output layer to complete the corresponding downstream tasks such as classification or prediction.

### 2.2. Deep-Learning-Based Graphic Processing Networks

#### 2.2.1. Faster R-CNN Network

Fast R-CNN [33] is a commonly used detection network framework that determines the compact enclosed borders of detected objects. Fast R-CNN integrates feature extraction, a region candidate network, target area pooling, and target classification into one network, thereby significantly improving the target detection speed. A region candidate network based on the Fast R-CNN detection framework is introduced to generate multiple candidate region reference frames rapidly, and normalized fixed-size region features are constructed for multi-size reference frames through the target region pooling layer. Using the shared convolutional network, the feature mapping is input to the region candidate network and target region pooling layer simultaneously, thereby reducing the number of parameters and computation of the convolutional layer.

#### 2.2.2. SSD Network

The Single Shot MultiBox Detector (SSD) [34], which is a fully convolutional target detection algorithm that was proposed in 2016, offers a significant speed advantage compared to the Faster R-CNN network. SSD is a one-stage algorithm that predicts the border and score of the object to be detected. The SSD algorithm uses a multi-scale approach in the detection process to construct multiple default frames on feature maps of different scales, and subsequently performs regression and classification. The final detection results are obtained using non-maximum suppression methods.

#### 2.2.3. CRNN Network

The convolutional RNN (CRNN) [35] is a prevalent graphical text recognition network that can recognize relatively long and variable text sequences. The feature extraction layer contains a CNN and bidirectional LSTM (BiLSTM), which can be trained end to end simultaneously. It learns the contextual relationship between character images using the BiLSTM and connectionist temporal classification (CTC) networks, which enhances the graphic recognition accuracy. Because the CRNN supports end-to-end combined training, the recognition network used in this study was also trained with the CRNN.

#### 2.2.4. RARE Network

The robust text recognizer with automatic rectification (RARE) [36] network is relatively effective in recognizing distorted image text. In the model inference process, the input image passes through the spatial transformation network to obtain the corrected image, which then enters the sequence recognition network to obtain the text prediction result.

## 3. Proposed Method

### 3.1. Pill Box Text Recognition

The recognition of pill box text requires two parts: text region detection and text recognition. Owing to the increased interest in neural networks over the past few years and the efficiency and accuracy that they provide, they have gradually replaced the traditional method of detecting natural scene text using a deep-learning-based approach to recognize pill box text. Comparisons with the traditional method have demonstrated that deep learning provides better recognition results and a faster recognition speed [37].

Two general methods are available for text detection: regression and segmentation methods. The Fast R-CNN and SSD models are the most common regression-based algorithms. In addition to their poor performance and lengthy training process, they cannot handle distorted shape data with poor shooting effects. DBNet uses an image segmentation method to detect arbitrary text shapes, for example, pill boxes. The process may include factors such as text occlusion, bending, and complex backgrounds.

Once the text region has been determined, the next step is to recognize the images in the region. Figure 1 depicts the traditional image detection and recognition processes. Traditional image detection and recognition methods are redundant and complex, with low recognition efficiency.

In this study, the DBNet detection framework was used in combination with a CRNN and CTC loss. Figure 2 presents the use of the DBNet-CRNN as the basis for end-to-end image text detection and recognition.

### 3.2. Text Region Detection

The text region detection module must locate the text region in the captured image. This method is used to draw a detection box that contains the text to provide the input for the subsequent text recognition. In this study, the DBNet algorithm was used to segment the data; compared with the more mainstream use of CTPN, DBNet can solve not only rotated and tilted text, but also distorted text. For the task of drug box text recognition, this method can greatly improve the accuracy of text detection. The structure of the DBNet model network is illustrated in Figure 3.

Figure 4 compares the traditional text detection algorithm and the DBNet detection algorithm. The main principle of the DBNet detection algorithm is the transformation of the probability map that is generated using segmentation methods into bounding boxes and text regions, which includes binarization postprocessing. Binarization is crucial in character recognition, and the traditional binarization operation by setting a fixed threshold is difficult to adapt to complex and variable detection scenarios (the blue process in the figure). In this study, the binarization operation was applied to the network and optimized simultaneously so that the threshold value of each pixel point could be predicted adaptively, that is, using DBNet (the red flow in the figure). Its text detector involves the insertion of differentiable binarization into the segmentation network as a simple segmentation network. In this manner, the adaptive thresholding of the entire heat map is eventually achieved.

The basic steps of differentiable binarization are as follows: First, the backbone and features of the input image are extracted. Subsequently, the image is passed to the feature giant, the same-size image is acquired during feature association, the acquired backbone and features are analyzed, and the predicted probability and threshold maps are calculated. Finally, the final approximate binarization map is obtained according to the predicted probability and threshold maps, and the text edge frame is generated. Standard binary processing is not differentiable and the segmentation network cannot be optimized during the training process. Therefore, microscopic binarization is used for improved computation of the inverse values.

Differentiable binarization differs from standard binarization in that the transitive function is approximated as follows:(1)$$\hat{B}=\frac{1}{1+e^{-k\left(P_{i,j}-T_{i,j}\right)}},\;$$
where $\hat{B}$ denotes the approximate binary map, *T* is the threshold feature map that is learned by the network, and *k* denotes the magnification, which is set to an empirical value of 50. This enables better distinction between the foreground and background.

The text detection header uses a feature pyramid network structure, which is divided into a bottom-up convolution operation and top-down sampling operation to obtain multi-scale features. The lower part of the structure is a 3 × 3 convolution operation, which obtains 1/2, 1/4, 1/8, 1/16, and 1/32 of the original image size according to the convolution formula. Subsequently, top-down up-sampling ×2 is performed, followed by fusion with the same size feature map that is generated from the bottom-up operation. After fusion, a 3 × 3 convolution is used to eliminate the blending effect of the up-sampling, and, finally, the output of each layer is rescaled and unified into a feature map of 1/4 size.

Thereafter, in the training phase, the probability map was determined by obtaining the binarized image. The binary map was then derived based on a threshold value and the connected region was determined. Finally, the text box was obtained by expanding and shrinking the sample region through the compensation calculation of Equation (2).
(2)$$D=\frac{A\left(1-r^{2}\right)}{L}$$

### 3.3. Text Recognition

The CRNN text recognition algorithm integrates feature extraction, sequence modeling, and transcription for natural scene text recognition. Its greatest advantage is that it can recognize text sequences of variable lengths, especially in natural scene text recognition tasks. Furthermore, the recognition accuracy is high after studying the surface.

Figure 5 depicts the CRNN recognition process, in which a convolutional layer, recurrent layer, and transcription layer are included. These layers are used for the feature extraction, sequence modeling, and transcription functions, respectively.

The feature sequences of the input image are first obtained in the convolutional layer and then converted into labels using per-frame prediction in the recurrent layer. The convolutional layer (Conv) contains Conv and MaxPool, which are used to transform the input image into a set of convolutional feature matrices, following which the BN network is used to normalize these convolutional feature matrices sequentially. Thereafter, the obtained feature vectors are arranged individually with the feature maps from left to right. The resulting feature sequence is illustrated in Figure 6.

As RNNs exhibit the problem of gradient disappearance and cannot obtain more contextual information, CRNN uses a BiLSTM structure in the recurrent layer. The label X = $x_{1}$,…… $x_{T}$ of the feature sequence $y_{i}$ is predicted for each frame $x_{q}$. Because the deep bidirectional recurrent network can memorize the sequence output of long segments, it can utilize the length of the time series in the persistent layer of the LSTM network to obtain more image information, and the forward and reverse directions of the BiLSTM network can obtain more complete image information, thereby effectively preventing the gradient disappearance problem. Figure 7 presents the structure of the deep bidirectional recurrent network.

The prediction result label $y_{i}$ of the loop layer is converted into a label sequence by the transcription layer at the top of the CRNN. Theoretically, the transcription layer uses the CTC algorithm to calculate the loss function, which simplifies the training process and causes it to converge rapidly. In the lexicon-free transcription mode, the label of the target text is mainly determined by the label with the highest probability among the predicted result labels. In the lexical transcription mode, the label of the target text is first determined by locating the target text and later by using a fuzzy matching algorithm.

The transcription layer indicates the label sequence $y_{i}$ using the predicted result label $y_{i}$ for each frame, and any result label $y_{i}$ therein is the set $L^{\prime }=L\cup${blank} on a probability distribution, as indicated in Equation (3).
(3)$$y_{t}\in {ℜ}^{\left|L^{\prime }\right|}$$

The mapping function β is defined on the sequence $\in {L^{\prime }}^{T}$, which is mainly used to eliminate blank or redundant labels. In this manner, the sequence $\pi \in {L^{\prime }}^{T}$ is written into sequence I, where T is the length and π is the predicted probability. For example, for the sequence “-hh-e-l-ll-0--” (where “-” is a blank character), the resultant sequence after removing blank characters and duplicate labels using the β mapping function is “hello.” The conditional probability is the sum of the probabilities of passing all sequences π through the β mapping function to label sequence I, as shown in Equation (4).
(4)$$p\left(I\;|\;y\right)=\underset{\pi :B\left(\pi \right)=I}{\sum }p\left(\pi \;|\;y\right)$$

The probability of *π* is defined as follows:
(5)$$p\left(\pi \;|\;y\right)=\prod_{t=1}^{T}\;{y_{\pi }^{t}}_{t},$$
where ${y_{\pi }^{t}}_{t}$ is the probability of representing label $\pi_{t}$ on timestamp *t*.

CRNN offers several advantages in the field of natural scene research. The CRNN model can be trained end to end and its training steps are easier than those of other algorithms. The CRNN model is not limited by the existence of dictionaries, and high accuracy and robustness can be obtained both with and without dictionaries. Furthermore, CRNN is not restricted by the length of the sequences and can handle arbitrary lengths. The CRNN model is smaller in overall size and requires fewer parameters, which provides better performance for many natural scene recognition applications.

## 4. Experiments

### 4.1. DBNet Model Training

#### Dataset and Experimental Setup

Dataset. The dataset used for training in this study was the text localization dataset from ICARD 2015, with a ratio of 2:1 between the training and test sets. The text files were comma-delimited files, where each line corresponds to a word in the image and provides its bounding box coordinates (four corners, clockwise) and its format for transcription. Anything after the eighth comma is part of the transcription and does not use escape characters. The “do not care” region is indicated by the transcription “####” in the basic facts. Some of the contents are displayed in Figure 8.

Training setup. The hardware environment that was used for the experiments was 12 GB RAM, a Xeon processor, Ubuntu ×64, and CUDA 11.7. The development environment was Python 3.6 and the compilation environment was PyTorch 1.4. The DBNet model was trained using a GPU, the initial learning rate was set to 0.001, and the neural network weights were updated iteratively based on the training data using the Adam optimization algorithm [22], with the epoch set to 1200 and the batch size set to 16. The intersection over union (IoU) of the DBNet was experimentally obtained, as illustrated in Figure 7.

As shown in Figure 9, the IoU value gradually increased from 0 to 0.6441. This indicates that the IoU value increased with an increase in the training time and eventually converged. The DBNet model exhibited a stable detection effect on the ICARD 2015 dataset, and the trained DBNet model was finally used for detection.

### 4.2. CRNN Model Training

#### Dataset and Experimental Setup

Dataset. The dataset used in this experiment was the Synthetic Chinese String Dataset, which is a Chinese recognition dataset that includes more than 3.6 million training images and 5824 characters. The ratio of the training and testing sets was 120:1. Specifically, the images were preprocessed, blurred, gray-scaled, and background-processed such that they were mostly black characters on a white background, which is ideal for training CRNNs. The text file included a dictionary and an index. The dictionary file contained 5989 common Chinese characters and a blank placeholder. Each Chinese character and placeholder occupied one line, totaling 5990 lines, as indicated in Figure 10a. In addition to the dictionary file, an index file was included, which contained the address of the training image, and the right-hand side indicated the position of the corresponding character in the dictionary, as illustrated in Figure 10b.

Training setup. The hardware environment for the experiment was 12 GB RAM, a Xeon processor, Ubuntu ×64, and CUDA 11.7. The development environment was Python 3.6, and the compilation environment was PyTorch 1.4. The CRNN model was implemented in the PyTorch framework, following which it was trained on the GPU, with the initial learning rate set to 0.001. The Adam optimization algorithm was used, the epochs were set to 16, and the batch size was 128. Finally, the CRNN could recognize objects in natural scenes. The detailed accuracy versus the loss value curve for the validation dataset is presented in Figure 11.

Figure 11 shows that as the number of training sessions increased, the accuracy of the model gradually increased, and greater accuracy was achieved. In contrast, the loss value varied with time and converged. This indicates that the training process gradually stabilized.

### 4.3. Pill Box Recognition and System Framework

The system framework for pill box detection and recognition is depicted in Figure 12. DBNet was used as the text detection module and CRNN was used as the text recognition module.

The system framework for pill box detection and recognition was designed based on B/S, which is an architecture that can significantly reduce the learning cost of users. As indicated in the figure, the image was first uploaded from the web side and stored in the database. Thereafter, the back-end read the image to be recognized from the database, predicted the angle of the text to be detected by the text orientation algorithm and processed it, recognized the text area in the image using the DBNet model and output the text box, stitched these text boxes together, and then sent them to the CRNN text detection model to recognize each line. Finally, the information was output to the web server.

### 4.4. Pill Box Recognition Experiment

Figure 13 shows the identification process for the pill box and the identification results.

A total of 100 pillboxes were used in the experiment. The results are presented in Table 1.

The experimental results demonstrate that the accuracy of the proposed method for locating medicine names from the pill boxes was as high as 93.2% and for recognizing the medicine names the accuracy was as high as 88.1%. The recognition accuracy of this method is similar to that of the other method because both use CRNN as the recognition algorithm; however, there is a significant difference in the localization accuracy. This also demonstrates that the DBNet segmentation algorithm offers a significant advantage in recognizing distorted and deformed text.

### 4.5. Drug Instruction Identification Experiment

During the experiment with drug box recognition, we found that the texts on drug boxes have large and small texts. Additionally, we found that the text content is not standard and even contains artistic words. Therefore, we designed a drug instruction manual recognition experiment to compare and analyze drug box recognition.

Figure 14 shows the identification process for the drug instruction and the identification results.

A total of 50 drug instructions were used in the experiment. The results are presented in Table 2.

From the analysis of the above experimental results, the results of the drug instruction manual recognition detection experiment significantly improved the accuracy rate when compared to the results of the drug box recognition detection experiment. It seems that the irregularity of the text on the drug box itself affects the accuracy rate of drug box content recognition detection. This is also due to the angle problem of the text itself.

## 5. Conclusions

We combined a DBNet detection network with a CRNN text recognition network to develop a pill box recognition method based on a deep learning approach. Unlike in other methods, prior image preprocessing was not required. The image could be input into the network directly, and, subsequently, output directly following automatic processing by the network. According to experiments using 100 pill boxes, the proposed method offers a significant advantage over other methods for recognizing medicine names.

In the experiments, it was difficult to achieve full recognition when locating the text on the pill box. This is because the pill box itself has many complex factors. Furthermore, although recognition of branch text could be achieved, it could not be output as a whole, which resulted in separate results. According to the results of testing on drug instruction manuals, the key factors affecting the accuracy of drug box content recognition detection was not only the angle problem, but also the irregularity of the text. The next step of research is to solve these two problems in order to improve the accuracy of drug name positioning.

“Smart medicine” is the future direction of medicine and the frontier of academia. As deep learning technology continues to break through, its performance in various fields is increasingly being discussed by academics. By integrating deep learning with medicine, the redundancy and complexity of the work of public health professionals can be reduced, and more powerful artificial intelligence can be used to assist them in completing related tasks more efficiently. The use of smarter, simpler and more er-ror-tolerant ways of entering and managing medicines will be an integral part of the future of medicine.

## Figures and Tables

**Figure 1 ijerph-20-03881-f001:**
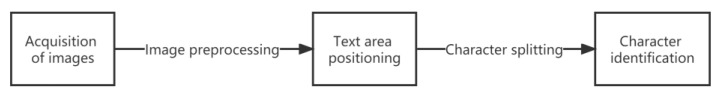
Traditional method for image text detection and recognition.

**Figure 2 ijerph-20-03881-f002:**
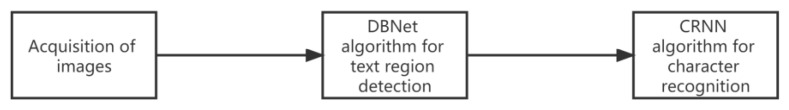
Process for image text detection and recognition based on DBNet-CRNN.

**Figure 3 ijerph-20-03881-f003:**
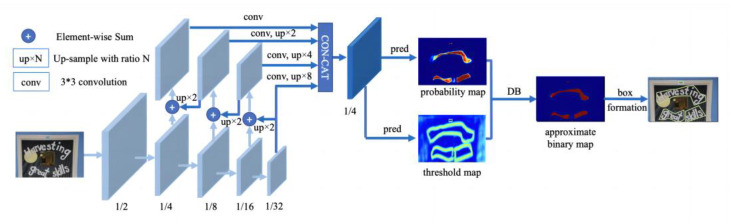
DBNet model network structure.

**Figure 4 ijerph-20-03881-f004:**
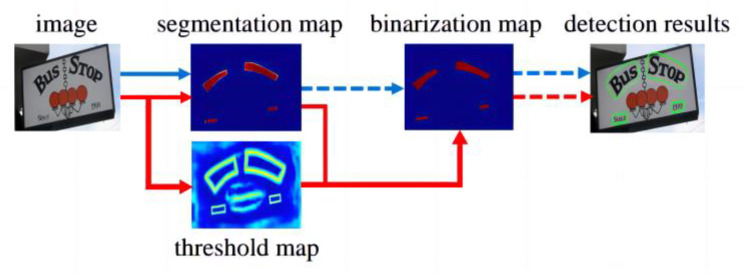
Comparison of traditional text detection algorithm and DBNet detection algorithm.

**Figure 5 ijerph-20-03881-f005:**
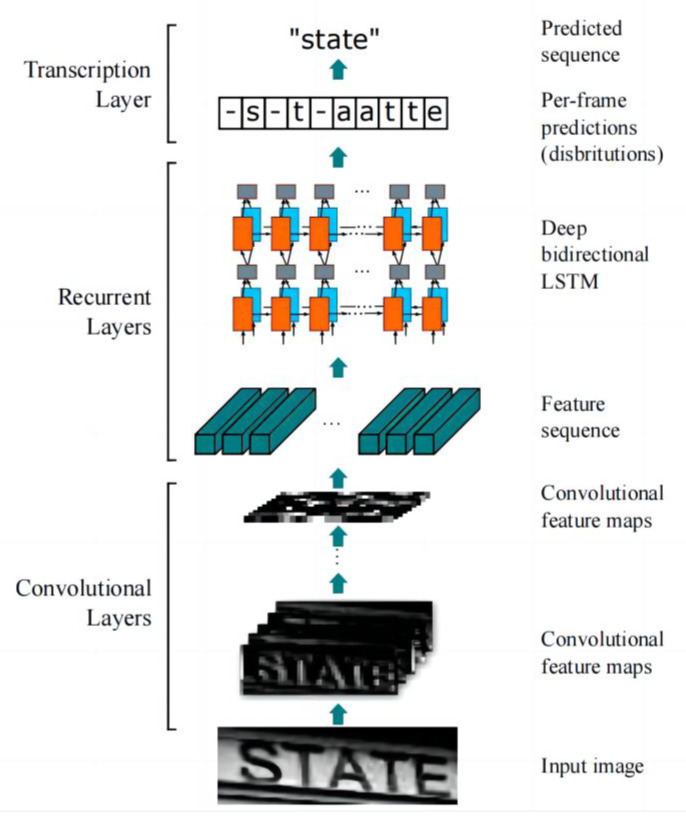
Flow diagram of CRNN algorithm.

**Figure 6 ijerph-20-03881-f006:**
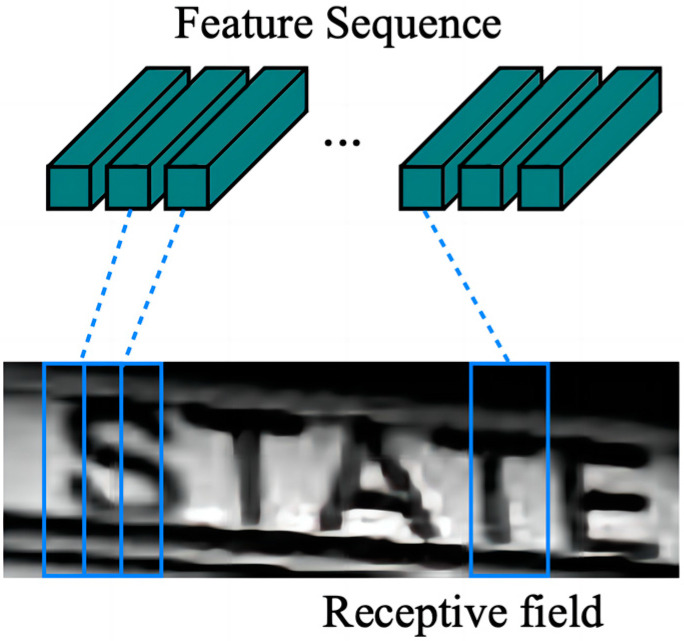
Receptive field. As each vector is associated with a receptive field on the input image, it can be considered as a feature vector of that field.

**Figure 7 ijerph-20-03881-f007:**
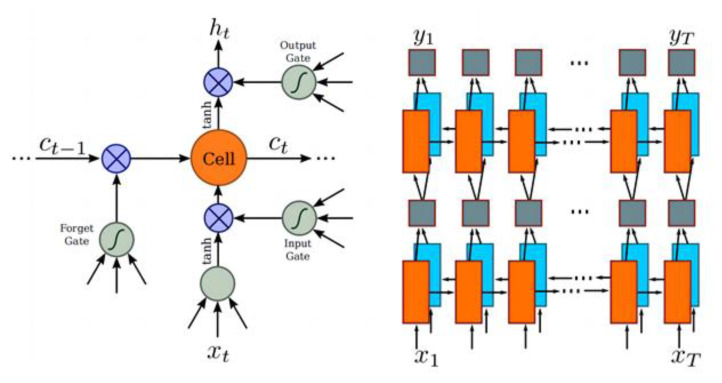
Deep bidirectional recurrent network structure.

**Figure 8 ijerph-20-03881-f008:**
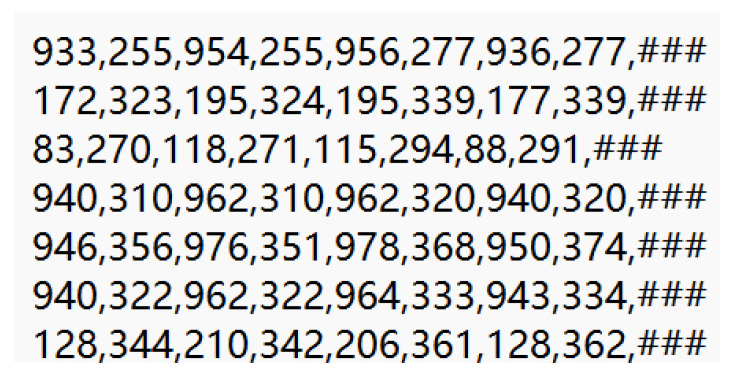
Text format tags used in DBNet dataset training.

**Figure 9 ijerph-20-03881-f009:**
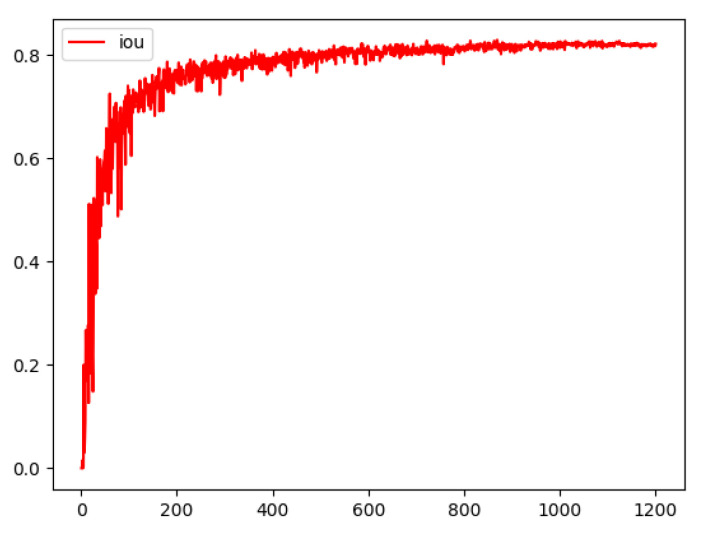
Trend graph of intersection over union (IoU) values in DBNet training.

**Figure 10 ijerph-20-03881-f010:**
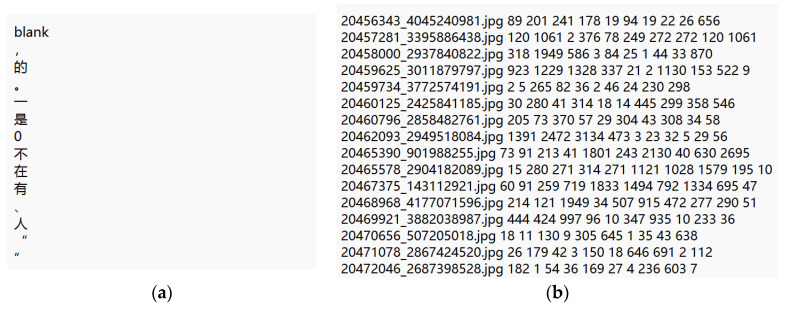
Data for CRNN training: (**a**) dictionary of characters and (**b**) index corresponding to image.

**Figure 11 ijerph-20-03881-f011:**
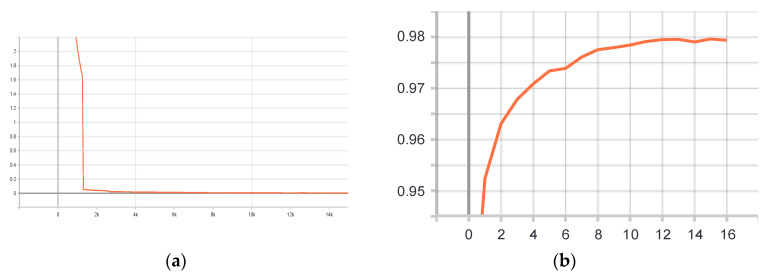
Training results of CRNN: (**a**) change in loss value and (**b**) variation in accuracy of validation datasets.

**Figure 12 ijerph-20-03881-f012:**
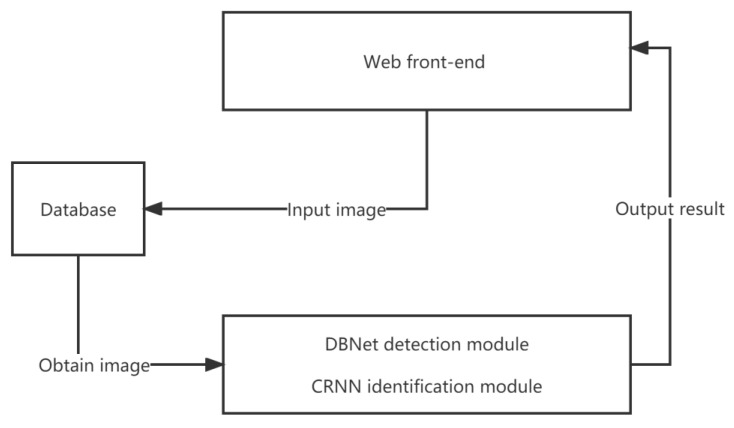
Flow of pill box text detection system.

**Figure 13 ijerph-20-03881-f013:**
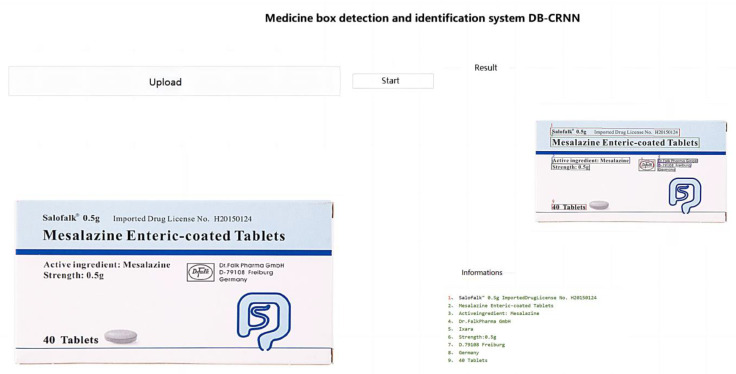
Pill box identification process and identification results.

**Figure 14 ijerph-20-03881-f014:**
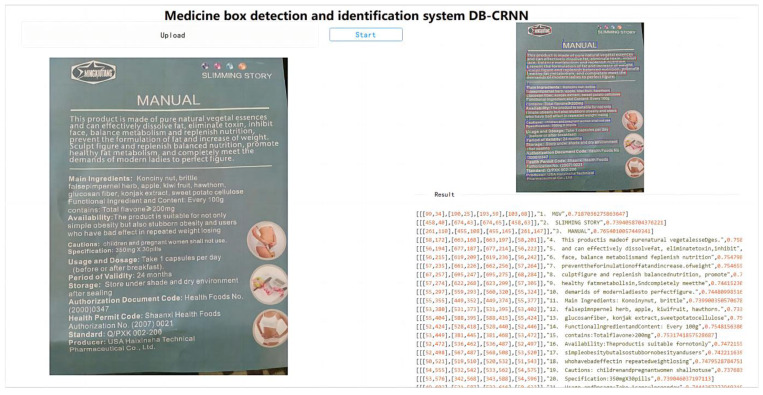
Drug instruction identification process and identification results.

**Table 1 ijerph-20-03881-t001:** Results of pill box identification using traditional and proposed methods.

Method	Sample Size	Text Location Accuracy	Text Recognition Accuracy
CTPN + CRNN	100	81.8%	87.9%
Ours	100	93.2%	88.1%

**Table 2 ijerph-20-03881-t002:** Results of drug instruction identification using traditional and proposed methods.

Method	Sample Size	Text Location Accuracy	Text Recognition Accuracy
CTPN + CRNN	50	94.2%	93.5%
Ours	50	96.7%	95.1%

## Data Availability

The data in this study are publicly available. The data were obtained from the Internet, but access is required by contacting the authors due to privacy or ethical concerns.

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
