# Peer review of "Pill Box Text Identification Using DBNet-CRNN"

_ijerph, 2023, doi:10.3390/ijerph20053881_

Round 1

Reviewer 1 Report

The authors proposed a paper titled “Pill Box Text Identification Using DBNet-CRNN”, which revolves around the issues related to the recognition of medicine names from the pill box images. The research topic is very important and the paper has an interesting findings that can be used in order to solve the problem of text recognition from images.

The following minor modifications are suggested to improve the quality of the paper.

Add the steps of the DBNet algorithm

Add the steps of CRNN

Figure 14  which shows the Drug instruction identification process and identification results, is not clear and not readable.

Author Response

Dear Editors and Reviewers:

Thank you for your letter and for the reviewers’ comments concerning our manuscript entitled “Pill Box Text Identification Using DBNet-CRNN” (ID: No. IJERPH-2188100). Those comments are all valuable and very helpful for revising and improving our paper, as well as the important guiding significance to our research. We have studied comments carefully and have made correction which we hope to meet with approval. Revised portions are marked in red in the paper. The main corrections in the paper and the responds to the reviewer’s comments are as flowing:

Responds to the reviewer’s comments:

Reviewer #1

Point #1 Add the steps of the DBNet algorithm.

Answer:Thanks for valuable comments,figure 3 shows the steps of the DBNet algorithm. Although the figure is only a DBNet model network structure diagram, it can also represent the DBNet algorithm steps.

Point #2 Add the steps of CRNN.

Answer:Thanks for valuable comments,figure 5 shows the steps of the CRNN algorithm. Although the figure is only a CRNN model flowchart, it can also represent the steps of the CRNN algorithm.

Point #3 Figure 14 which shows the Drug instruction identification process and identification results, is not clear and not readable.

Answer:We apologize that the image we provided was not very clear and we have replaced it to make the content more readable.

Reviewer 2 Report

This paper aims to combined a DBNet detection network with a CRNN text recognition network to develop a pill box recognition method. This is a good research topic. However, there are still many places to be modified. 

Introduction Part:

1The paper only introduces that the preprocessing of traditional OCR is gradually being replaced with deep learning with simple steps. But the related references are lack.

2Next the paper introduces the preprocessing process, such as image segmentationimage rotation correctionline detectionimage matchingtext outline extraction and local segmentation. It also introduces several algorithms used in each step. It does not explain which method you should use in your article and why you should use your solution for the "detection and recognition of pill box text".

Related work:

Just explained the deep learning and several deep learning algorithms, and did not write about the research work related to "detection and recognition of pill box text", and the differences between your research and related research. Are there other studies using relevant algorithms to use pill box text identification? 

Proposed Method

Didn't explain the difference between your algorithm and CTPN+CRNN algorithm.

Experiments

1Only 4.1.1 Dataset and Experimental Setup, not 4.1.2

2Only 4.14.24.4, not 4.3

Author Response

Dear Editors and Reviewers:

Thank you for your letter and for the reviewers’ comments concerning our manuscript entitled “Pill Box Text Identification Using DBNet-CRNN” (ID: No. IJERPH-2188100). Those comments are all valuable and very helpful for revising and improving our paper, as well as the important guiding significance to our research. We have studied comments carefully and have made correction which we hope to meet with approval. Revised portions are marked in red in the paper. The main corrections in the paper and the responds to the reviewer’s comments are as flowing:

Responds to the reviewer’s comments:

Reviewer #2

Point #1 The paper only introduces that the preprocessing of traditional OCR is gradually being replaced with deep learning with simple steps. But the related references are lack.

Answer:Thanks for valuable comments, we have added relevant references[26] to support our view.

Point #2 Next the paper introduces the preprocessing process, such as image segmentation、image rotation correction、line detection、image matching、text outline extraction and local segmentation. It also introduces several algorithms used in each step. It does not explain which method you should use in your article and why you should use your solution for the "detection and recognition of pill box text".

Answer:Thanks for valuable comments, we have added relevant references to support the excellence of end-to-end networks for text recognition. We have also added relevant content to explain why an end-to-end approach is used for the recognition of pillbox text. in section 4, add “Xuebo Liu et al. proposed an end-to-end network for FOTS, which saves time and learns more image features than the traditional approach [27]. Christian Bartz et al. proposed an STN-OCR end-to-end network, which improves the accuracy of the recognition phase by embedding a spatial transformation network in the detection process to perform affine transformation on the input image [28]. Siyang Qin et al. proposed Unconstrained end-to-end network to simplify the recognition process by reducing the detection of arbi-trarily shaped text to an instance segmentation problem without the need to do text region correction in advance[29].

With the implementation of information technology in pharmaceutical systems, more and more complex and redundant tasks can be gradually improved by means of in-formation technology. Among them, the entry of drug box information is a time-consuming and labor-intensive task that is also error-prone. Based on the excellent recognition capabilities of end-to-end deep learning networks, they can be used to reduce the complexity of this task while improving the accuracy of entry. In this paper, we pro-pose and design a deep learning based end-to-end scene text recognition algorithm for the task of recognizing pill box text.

Point #3 Just explained the deep learning and several deep learning algorithms, and did not write about the research work related to "detection and recognition of pill box text", and the differences between your research and related research. Are there other studies using relevant algorithms to use pill box text identification?

Answer:Thanks for valuable comments, there is no further research on the use of deep learning for the recognition of pillbox text. Our end-to-end approach based on deep learning is better than the traditional way of text recognition. The related literature is described in the add section of Section 1.

Point #4 Didn't explain the difference between your algorithm and CTPN+CRNN algorithm.

Answer:We apologize for the oversight in the introduction of this content. in section 3.2, add “compared with the more mainstream use of CTPN, DBNet can solve not only rotated and tilted text, but also distorted text, for the task of drug box text recognition, this way can greatly improve the accuracy of text detection.

Point #5 Only 4.1.1 Dataset and Experimental Setup, not 4.1.2

Answer:We are very sorry for our negligence of  presentations, we have added the relevant serial numbers.

Point #5 Only 4.1、4.2、4.4, not 4.3

Answer:We are very sorry for our negligence of  presentations, we have readjusted the relevant serial numbers.